# Probiotic Monotherapy with *Lactobacillus reuteri* (Prodentis) as a Coadjutant to Reduce Subgingival Dysbiosis in a Patient with Periodontitis

**DOI:** 10.3390/ijerph19137835

**Published:** 2022-06-26

**Authors:** Claudia Salinas-Azuceno, Miryam Martínez-Hernández, José-Isaac Maldonado-Noriega, Adriana-Patricia Rodríguez-Hernández, Laurie-Ann Ximenez-Fyvie

**Affiliations:** 1Laboratorio de Genética Molecular, Facultad de Odontología, Universidad Nacional Autónoma de México (UNAM), Ciudad de México 04360, Mexico; claous_02@comunidad.unam.mx (C.S.-A.); miryam_mh@comunidad.unam.mx (M.M.-H.); lximenez@post.harvard.edu (L.-A.X.-F.); 2Centro de Estudios Superiores de Ortodoncia (CESO), Nicolás San Juan 1628, Ciudad de México 03100, Mexico; drisaac.maldonado@gmail.com

**Keywords:** oral probiotics, *Lactobacillus reuteri* Prodentis, periodontitis, DNA–DNA hybridization, subgingival dysbiosis

## Abstract

(1) Background: Probiotics can be considered a non-invasive periodontal monotherapy for the modulation of microbiota when periodontal treatment is not accessible. The aim was to evaluate the ability of *Lactobacillus reuteri* Prodentis as monotherapy to modulate periodontal parameters and subgingival biofilm dysbiosis. (2) Methods: A 30-year-old patient with periodontitis was followed longitudinally after one month of daily consumption of *L. reuteri* Prodentis (T0). Periodontal measurements and microbial identification by Checkerboard DNA–DNA hybridization of 40 bacteria were compared between baseline (T0) and 30 days (T1) or 90 days (T2), using the Kruskal–Wallis (KW) and Mann–Whitney U (MW) tests. (3) Results: Low values of pocket depth, attachment level, dental plaque, gingival erythema (GE), and suppuration were observed at T0 vs. T1, with the clinical improvement of GE (*p <* 0.05, MW) and the recovery of tooth 46 fistulation. T1 vs. T0 comparisons showed lower levels (Lev) or proportions (Prop) of *Parvimonas micra* (Lev: *p <* 0.05, MW; Prop: *p <* 0.01, MW) and *Streptococcus gordonii* (Prop: *p <* 0.05, MW), and a predominance (Lev/Prop) of *Actinomyces odontolyticus* and *Streptococcus mitis*; lower levels and proportions of *P. micra*, *Eubacterium saburreum*, *Porphyromonas gingivalis*, and *Tannerella forsythia* were observed in tooth 46 (T1/T2 vs. T0). (4) Conclusions: Under monotherapy with *L. reuteri* Prodentis, periodontal measurements of the patient were maintained, with selective changes in the subgingival microbiota that were proportional to the time of probiotic administration, with any additional periodontal treatment.

## 1. Introduction

The oral cavity is the window of the body, and it is home to a diverse microbial community encompassing one of the most complex microbiomes and dynamic microbial communities, comprising hundreds of different species of bacteria living in a structure known as a biofilm. Dysbiosis of the periodontal microbiota can interfere with the normal function of the host immune system, resulting in the development of inflammatory diseases, such as periodontal disease [1,2,3]. Periodontitis is an infectious and inflammatory disorder that is characterized by the destruction of the supporting structure of teeth, induced by biofilm dysbiosis, in which a higher proportion of major pathogens such as red complex species, including *Porphyromonas gingivalis*, *Tannerella forsythia*, and *Treponema denticola*, and a low proportion of beneficial strains, such as *Actinomyces* sp. or *Streptococcus* sp., remain the key to microbial disruption [3,4,5]. Despite improvements in preventive measures over the past 40 years, periodontal disease continues to contribute to widespread oral health dysfunction and an increased susceptibility to systemic diseases. Chronic unresolved hyperinflammation during periodontal disease is strongly associated with systemic conditions such as diabetes and obesity, and cardiovascular and neurological diseases, and is caused by the dysbiosis of the oral microbiome; therefore, major periodontal pathogens may be the targets of therapeutic interventions, as has recently been reported in the literature [6,7]. Periodontitis is considered to be the sixth most prevalent disease worldwide, with an estimated 743 million people affected, and with individuals in low- and middle-income countries being the most commonly affected [8].

Procedures aimed at dysbiotic biofilm control are essential for the treatment of periodontal diseases, with supragingival plaque control, together with scaling and root planning (SRP), being the main forms of periodontitis treatment [9]. However, despite its clinical benefits [9], SRP therapy is not always an available treatment among the general population, especially for hospitalized or disabled subjects, or for those who cannot afford the therapy. In these cases, where plaque control or SRP is not possible, alternatives to modulate the oral microbiome by other means, such as the introduction of live microbial strains, or so-called probiotics, could be considered as encouraging and non-invasive alternatives for modulating virulence-associated oral microbiome dysbiosis.

Over the past 20 years, several probiotic products have been marked for the modulation of the human gut microbiome, but only a few have been proposed specifically for modulation of the oral microbiome [10,11]. Among them, the probiotic *L. reuteri* is beneficial in improving the clinical and microbiological parameters that are associated with periodontitis, by resolving inflammation and by reducing the molecular mediators associated with bone loss [12,13,14,15]. The probiotic *L. reuteri* has been recognized as a homeostasis enhancer by promoting beneficial subgingival strains on major periodontal pathogens [16,17]. In addition, and given that contradictory results refute the inflammatory or microbiological benefits of *L. reuteri* [17,18], the present case report represents a research gap supporting monotherapy for the management of periodontitis, with the advantage of evaluating microbial levels and proportions with the Checkerboard technique of 40 representative periodontal strains.

Since periodontal clinical parameters and subgingival microbiological changes under probiotic treatment have rarely been evaluated, the purpose of the present study was to evaluate the ability of the probiotic *Lactobacillus reuteri* Prodentis as monotherapy on subgingival biofilm. The hypothesis posed that the oral probiotic monitored longitudinally after one month of daily consumption, at one and three months of follow-up of a patient with generalized periodontitis, would modulate their clinical periodontal parameters and the subgingival biofilm dysbiosis associated with the presence of periodontal disease.

## 2. Materials and Methods

### 2.1. Case Report

The procedures followed were under the regulations of the corresponding clinical research ethics committee and those of the Code of Ethics of the World Medical Association (Declaration of Helsinki), and they were developed by the Molecular Genetics Laboratory of the School of Dentistry, with ISO:9001:2015 quality management system certification from UNAM. The clinical practices have been developed according to the universal regulations of informed consent, and the FDA approves BioGaia’s Probiotic designation. The clinical measurements and microbial sampling are under the approval of the Molecular Genetics protocols by the acceptance of the corresponding Ethics Committee (CorInv/204/2019).

A 30-year-old female patient with periodontitis was monitored longitudinally for up to 60 days after one month of daily consumption of the probiotic *Lactobacillus reuteri* Prodentis. Clinical periodontal measurements and subgingival microbial identification were performed at baseline (T0), immediately after one month of oral probiotics consumption (T1, 30 days after baseline), and then at two months (T2, 90 days after baseline). The patient gave informed consent at each time point of the evaluation (T0, T1, and T2), acknowledging their willingness to participate in the present study.

### 2.2. Clinical Case Description and Probiotic Adjuvant

The patient reported having no systemic diseases, was not pregnant or nursing, was Mexican by birth origin and by direct Mexican descendants, had not consumed antibiotics within three months of the study, and had reported not being a current smoker and not having received previous periodontal treatment. She presented more than 20 natural teeth for clinical and microbial evaluation.

The patient received *L. reuteri* Prodentis (a patented lactic acid bacteria). The lozenges contained dried live *L. reuteri* 1 × 10^8^ CFU (DSM179381) cells plus *L. reuteri* 1 × 10^8^ CFU (ATCC PTA 5289), suspended in a mixture of fully hydrogenated palm oil, peppermint flavor, menthol flavor, peppermint oil, and sucralose. Each lozenge was composed of a minimum of 200 million live *L. reuteri* Prodentis. The patient was instructed to orally dissolve one probiotic tablet twice daily, once in the morning and once in the evening after toothbrushing, for 30 days. The patient received no professional prophylaxis or instruction on toothbrushing and was instructed to avoid any additional oral antiseptic solution or flossing in addition to her usual daily toothbrush.

### 2.3. Clinical Periodontal Evaluation and Microbial Assessment

Periodontal parameters included previous periodontal treatment, the number of missing teeth, and clinical measurements taken at six sites per tooth of all teeth, excluding the third molars (168 sites minimum). Periodontal measurements assessed the pocket depth, PD (mm), and the clinical attachment level, CAL (mm), recorded twice to the nearest millimeter by the same examiner, using a North Carolina periodontal probe (15 mm), and the mean of the pair of measurements was used for analyses. Plaque accumulation (PLA, 0/1: not detected/detected, non-revealing), gingival erythema (GE, 0/1), bleeding on probing, BOP (0/1, at the probing measurements), and suppuration, SUP (0/1, at the probing measurements) were as described in the literature [19]. The patient had been diagnosed according to the latest classification scheme for periodontal diseases and conditions [4], using the data obtained at the initial evaluation (T0).

Periodontal clinical data and subgingival biofilm samples were collected in a single appointment by a previously calibrated clinician. Subgingival biofilm samples were obtained as follows: After drying and isolation with cotton rolls, the supragingival plaque was removed with curettes, and subgingival plaque samples were obtained with sterile Gracey 11/12 curettes from the mesiobuccal site of all teeth present (excluding the third molars). Samples were placed in individual tubes containing 150 µL of TE buffer (10 mM Tris-HCl, 0.1 mM EDTA, pH 7.6), dispersed, and 100 µL of 0.5 M NaOH was added to each tube for further microbial evaluation.

Digoxigenin-labelled whole-genomic DNA probes were prepared using a random primer technique. Subgingival biofilm samples were individually processed for the identification of 40 microbial species using the Checkerboard DNA–DNA hybridization technique, following the procedures previously described [19]. The list of oral bacterial strains employed for the preparation of DNA probes and used in the present study is presented in Table 1. DNA was isolated and purified from lyophilized stocks of the American Type Culture Collection (Rockville, MD, USA). The specificity and sensitivity of the DNA probes were evaluated, and the sensitivity of the assay was adjusted to approximately 10^4^ cells for each species. Subgingival plaque samples were processed to determine the levels and proportions of the 40 microbial species, as previously described [19].

### 2.4. Statistical Analysis

All data are expressed as mean ± standard error of the mean (SEM). Differences in clinical parameters were sought between different evaluation times (T0, T1, and T2), using the Kruskal–Wallis test, and between paired comparisons of T0 vs. T1 or T0 vs. T2, assessed using the Mann–Whitney U (MW) test. Microbial data were obtained from the absolute counts of each of the 40 test species from up to 28 patient subgingival plaque samples from the patient. The analyses compared subgingival plaque composition between the different time points assessed (T0, T1, and T2), expressed as the mean levels of total and individual species (DNA probe count), and the mean proportions of individual species and microbial complexes (% total DNA probe count). The proportions were also determined by grouping the test species as being as similar as possible to previous descriptions of microbial complexes in the subgingival plaque, according to color complex [5]. Exceptions are listed in Table 1. Each data set was computed for individual species, and the microbial complexes for each sample were averaged within the subject, and then at each of the different evaluation times (T0, T1, and T2). Microbial differences were sought between paired comparisons of T0 vs. T1, or T0 vs. T2, as assessed using the KW and the MW test. For all of the statistical tests, significance values were obtained in SPSS software and adjusted for multiple comparisons, as previously described [19].

## 3. Results

### 3.1. Clinical Periodontal Characteristics

At the baseline, the patient had 25 teeth and had not undergone any tooth extraction during the follow-up longitudinal period. In addition to the upper third molars, the observed missing teeth corresponded to the upper first molars, the upper central and lateral incisors, and the lower-left incisor (Appendix A).

Special attention was paid to the right mandibular first molar (numbered 46, according to FDI nomenclature), in which an active periradicular infection with vestibular fistulation and radiography bone loss (4 sites with AL ≥ 5 mm, T0) due to an endodontic failure was observed during the baseline evaluation; thus, it was decided that an exploration of the microbial and clinical changes of this individual tooth would be included throughout the follow-up period. The fistulation presented in tooth 46 at T0 recovered completely after probiotic consumption, as observed in the evaluation of T1 and T2 (Appendix A). The periodontal clinical characteristics of full mouth and tooth 46 are presented in Table 2.

Based on the clinical periodontal evaluation at baseline (T0), the patient was diagnosed with generalized periodontitis, stage IV, grade B [4], with a mean PD of 3.9 ± 0.2 mm, an AL of 2.0 ± 0.2, and eight sites with AL ≥ 5 mm (Table 2 and Appendix A). After probiotic consumption, the clinical periodontal characteristics of the full mouth showed clinical improvement. As can be observed, the lower values of PD (T1: 3.3 mm, T2: 3.4 mm), AL (T1: 1.6 mm, T2: 2 mm), and the number of sites with AL ≥ 5 mm (T1: 4 sites, T2: 6 sites), PLA (T1: 44.2 %), GE (T1: 58 %), and SUP (T1: 0.7 %) were detected. Additionally, PLA (T2: 69.6 %, *p <* 0.01 UM, *p <* 0.05 KW), GE (T2: 98.6 %, *p <* 0.01 UM), and BOP (T2: 66.7 %, *p <* 0.05 MW) were clinical parameters that showed a significantly higher value at T2, compared to T0 evaluation (Table 2). Additional data included the patient’s perception at the follow-up of T1 and T2, which was an evident reduction in halitosis and bleeding gums after one month of consumption of *L. reuteri* Prodentis. On the other hand, tooth 46 exhibited clinical improvement in all periodontal clinical parameters at T1. GE was the clinical feature that showed the greatest reduction after probiotic consumption (T0: 100% vs. T1: 16.7%, *p <* 0.05 MW) (Table 2).

### 3.2. Mean Total Levels of Full Mouth and Tooth 46

Figure 1 shows the mean total levels (total DNA probe count × 10^5^ ± SEM) of the microorganisms in each follow-up evaluation of the full mouth and tooth 46. As observed, in the full mouth evaluation, there was a non-significant increase in the mean total levels after 30 days of probiotic consumption, with levels of 138.5 ± 23.9% at T1, and higher levels at evaluation after 90 days (T2: 204.9 ± 31.9%) compared to the baseline (T0: 115.1 ± 17.7%) (Figure 1a).

Regarding the mean total levels of tooth 46, it was observed that these were constant in the follow-up evaluations, with a slight increase being observed at T1 and T2 compared to the T0 evaluation (T1: 66.7 ± 3.0% and T3: 63.3 ± 2.8% vs. T0: 55.0 ± 2.7%) (Figure 1b).

### 3.3. Mean Individual Levels and Proportion of Full Mouth and Tooth 46

Figure 2 shows the mean individual total levels (total DNA probe count × 10^5^) and the mean proportions (% total DNA probe count) of the 40 species detected individually in the full mouth and tooth 46 subgingival plaque samples at each follow-up evaluation (T0, T1, and T2).

The full mouth microbial findings showed significant differences for nine bacterial species during the comparison of the T0 vs. T1 and T0 vs. T2 evaluation times (the KW and MW tests) (Figure 2a). When the T0 evaluation was compared to T1, significantly lower levels of *Parvimonas micra* (T0: 8.5 ± 0.7 vs. T1: 4.3 ± 0.8, *p <* 0.05 MW), and significantly higher levels of *Prevotella loescheii* (T0: 0.7 ± 0.5 vs. T1: 4.3 ± 0.9, *p <* 0.01 MW) and *Gemella morbillorum* (T0: 0.6 ± 0.3 vs. T1: 2 ± 0.5, *p <* 0.01 KW) were found. On the other hand, significantly higher levels of beneficial species, such as *Actinomyces odontolyticus* (T0: 1.6 ± 0.5 vs. T2: 5.9 ± 1.1, *p <* 0.01 MW), *Veillonella parvula* (T0: 5 ± 1.0 vs. T2: 11.4 ± 1.9, *p <* 0.05 KW), *Campylobacter showae* (T0: 0.8 ± 0.3 vs. T2: 4.5 ± 0.9, *p <* 0.05 MW), and *Capnocytophaga sputigena* (T0: 0.8 ± 0.3 vs. T2: 5 ± 1, *p <* 0.05 MW); and pathogenic species, such as *Fusobacterium periodonticum* (T0: 1.6 ± 0.5 vs. T2: 9.3 ± 1.3, *p <* 0.001 MW) and *T. denticola* (T0: 2.9 ± 0.8 vs. T2: 10.4 ± 1.7, *p <* 0.05 MW) were found in the T2 assessment, compared to the initial evaluation (T0).

Regarding the microbial findings of tooth 46, no significant differences were found between the T0, T1, and T2 evaluations (the KW or MW tests) (Figure 2b). However, in the T1 and T2 evaluations, a predominance in the levels of beneficial species, such as *A. odontolyticus* (T0: 0 vs. T1: 5 and T2: 4) and *Streptococcus mitis* (T0: 0 vs. T1: 0.58 or T2: 1.3); and lower levels of the putative pathogenic species, *P. micra* (T0: 4.5 vs. T1: 1.7, and T2: 0), *Eubacterium saburreum* (T0: 1.73 vs. T1 and T2: 0), and the red complex pathogenic species *P. gingivalis* (T0: 8.7 vs. T1: 1.6, and T2: 2.6) and *T. forsythia* (T0: 7.0 vs. T1: 0, T2: 0.6) were observed.

On the other hand, when the mean proportions of the 40 individual species in the full mouth subgingival plaque samples were analyzed between the T0 and T1 assessments (Figure 2c), a significantly lower percentage of the species *Streptococcus gordonii* (T0: 4.8 ± 0.8 vs. T1: 1.1 ± 0.2, *p <* 0.05 MW) and *P. micra* (T0: 9.5 ± 1 vs. T1: 4.1 ± 0.6, *p <* 0.01 MW), and higher proportions of beneficial species such as *Actinomyces naeslundii* (T0: 4.6 ± 1.2 vs. T1: 6.1 ± 1.8, *p <* 0.01 KW) and *G. morbillorum* (T0: 0.4 ± 0.1 vs. T1: 1.6 ± 0.4, *p <* 0.05 MW and *p <* 0.001 KW), and putative pathogenic species such as *P. loescheii* (T0: 0.2 ± 0.2 vs. T1: 2.6 ± 0.6, *p <* 0.01 MW) and *F. periodonticum* (T0: 0.8 ± 0.2 vs. T1: 2.9 ± 0.5, *p <* 0.01 MW) were found. When the T0 and T2 evaluations were compared, in the T2 evaluation, there were significantly lower proportions of pathogenic species such as *E. saburreum* (T0: 5.4 ± 0.6 vs. T2: 1.9 ± 0.4, *p <* 0.001 MW) and *Propionibacterium acnes* (T0: 2.1 ± 0.3 vs. T2: 0.5 ± 0.1, *p <* 0.01 MW), and higher proportions of beneficial species such as *A. odontolyticus* (T0: 0.8 ± 1.6 vs. T2: 4.8 ± 1.4, *p <* 0.001 MW), *C. showae* (T0: 0.4 ± 0.1 vs. T2: 1.6 ± 0.2, *p <* 0.01 MW), and *C. sputigena* (T0: 0.5 ± 0.2 vs. T2: 1.9 ± 0.4, *p <* 0.01 KW); and of pathogenic species such as *F. periodonticum* (T0: 0.8 ± 0.2 vs. T2: 5.2 ± 0.6, *p <* 0.001 MW) and *T. denticola* (T0: 2.2 ± 0.5 vs. T2: 5.1 ± 0.5, *p <* 0.05 MW).

Finally, regarding the mean proportion of subgingival bacterial species identified on tooth 46, no significant differences were found between the T0, T1, and T2 evaluations (Figure 2d). However, similar to the mean level data observed, the beneficial species *A. odontolyticus* (T1: 7.4 and T2: 6.3 vs. T0: 0) and *S. mitis* (T1: 0.9 and T2: 2 vs. T0: 0) showed a higher percentage in the T1 and T2 evaluations compared to the T0 evaluation, in addition to a lower proportion of pathogenic species such as *P. micra* (T1: 2.6, and T2: 0 vs. T0: 8.2), *E. saburreum* (T1 and T2: 0 vs. T0: 3.1,), and red complex pathogenic species *P. gingivalis* (T1: 2.5 and T2: 4.1 vs. T0: 15.8) and *T. forsythia* (T1: 0 and T2: 1 vs. T0: 12.9).

### 3.4. Mean Proportions of Microbial Complexes

The mean proportions (% of the total DNA probe count) of microbial complexes from full mouth and tooth 46 subgingival plaque samples at baseline (T0) and at the T2 and T3 evaluations are summarized in Figure 3. In the full mouth evaluation, the purple complex showed the highest significant increase at T1 compared to the baseline (T0) evaluation (T1: 11.1 ± 1.5 vs. T0: 4.8 ± 0.9, *p <* 0.001 KW and *p <* 0.01 MW). The rest of the complexes remained constant throughout the follow-up period; however, it was observed that the orange, red, and ungrouped complexes showed lower proportions in addition to the highest proportions of the periodontally beneficial *Actinomyces*, purple, and green strains at the T1 evaluation (immediately after probiotic consumption). On the other hand, the subgingival microbiota of tooth 46 showed an evident reduction in the red complex (periodontal pathogens) at the T1 and T2 evaluations vs. baseline (red complex: T1: 5.2, and T2: 7.1 vs. T0: 28.7%), while putative species (orange complex) showed the highest pie chart proportions in the T1 and T2 assessments versus baseline (T1: 56.6, and T2: 41.3 vs. T0: 13.0%).

## 4. Discussion

For a long time, the consumption of probiotics has been shown to be effective in the treatment of commensal infections such as ulcerative colitis, irritable colon, or antibiotic-associated diarrhea [20,21]. However, the use of probiotics as monotherapy for oral infectious diseases has barely been explored.

*L. reuteri* Prodentis is a commensal strain in the oral cavity and has the property of inhibiting bacterial growth through its reuterina protein [17]. The efficacy of *L. reuteri* as an intestinal or oral probiotic against commensal pathogens is based on its low pH tolerance, adhesion, and the antimicrobial properties related to its peptides [17,21,22]. Therefore, the present study aimed to explore the use of the *L. reuteri* Prodentis strain as monotherapy for modulating the clinical periodontal parameters and subgingival microbial dysbiosis in a subject with periodontitis.

Few studies have tested probiotics as monotherapy for the treatment of periodontitis [14,23]. Vicario et al. [14] conducted a trial evaluating the clinical effect of *L. reuteri* Prodentis administration as monotherapy for the treatment of initial to moderate periodontitis, and observed that plaque accumulation, bleeding upon probing, and pocket probing depths were significantly reduced after the 30-day probiotic intervention. Although no significant differences were found in the present study in the values of pocket probing depths in the follow-up period, a significant reduction in GE and BOP was observed at T1 of the full mouth evaluation, and specifically at tooth 46, where in addition to GE, suppuration disappeared at the T1 evaluation as well.

In addition, in the present study, it was considered appropriate to include a complete individual analysis of tooth 46, since at the baseline evaluation (T0), it presented a periradicular infection with vestibular fistulation due to endodontic failure. It is established that acute lesions in the periodontium can result in the rapid destruction of the periodontium, with a negative impact on the prognosis of the affected tooth, which under certain circumstances can have severe systemic consequences [24]. In this case report, a significant reduction in gingival erythema was observed in tooth 46 at T1 evaluation, and in addition to a slight reduction in pocket depth, AL, and plaque accumulation, these findings were associated with fistula regression in tooth 46 when the probiotic *L. reuteri* was employed as monotherapy.

In general, previous reports on the efficacy of *L. reuteri* Prodentis in modulating oral biofilm composition have focused on the evaluation of a few periodontal pathogenic bacteria [16,25,26]. However, a more complex and varied diversity of beneficial and pathogenic species is involved in the pathogenesis of periodontal diseases. It should be noted that an advantage of using the Checkerboard technique, as in the present study, is the possibility for enumerating 40 bacterial species in microbiologically complex systems, and provision of the quantification of representative strains that are strongly related to the pathogenesis of periodontitis is crucial for a periodontal microbial diagnosis [27,28]. Sinkiewicz et al. evaluated the microbial changes of dental plaque bacteria using the Checkerboard technique, and found that after the consumption of *L. reuteri* Prodentis through chewing gum, no significant differences were found between the subgingival strains of patients who had consumed the oral probiotics for 12 weeks and those of patients who consumed a placebo [17,22]. The findings of Sinkiewicz et al. agree with the data shown in the present case report, in which total bacterial counts showed no significant differences between all evaluations (T0, T1, and T2). Therefore, it can be assumed that not only was a correlation achieved between the maintenance of periodontal status and subgingival microbiota during probiotic consumption, but it can also be hypothesized that there was a presence of a temporary masking probiotic effect on selected strains, and not on the total amount of subgingival plaque [16,17]. In the present study, both in the full mouth and in one of the teeth that was most greatly affected by periodontitis (number 46), the total counts remained significantly unchanged after one month of *L. reuteri* Prodentis consumption by the patient. Sinkiewicz et al. [17] also reported that an increase in the mean score of the subgingival strains is dominated by an increase in beneficial strains such as *Actinomyces* sp. and *Streptococcus* sp. On the other hand, the main pathogenic or putative strains, such as *P. gingivalis*, *T. forsythia*, or *P. micra* were reduced after probiotic consumption, a finding that was also observed in the present case after 30 days of probiotic consumption, versus approximately 84 days of probiotic use in other similar reported studies [17,25].

Other studies have reported on the efficacy of probiotic *L. reuteri* combined with SRP treatment, and they have reported reductions in the pathogenic strains *P. gingivalis*, *P. intermedia*, or *A. actinomycetemcomitans*, compared to the placebo [25,26]; these findings are in agreement with the data shown in the present case report, where the pathogenic strains *P. gingivalis* and *T. forsythia* showed reduced proportions after consuming the probiotic adjutant, despite any additional mechanical periodontal treatment.

In the T2 full mouth evaluation, significantly higher levels or proportions of putative orange complex strains were observed compared to the baseline or T1, which would suggest that the probiotic effect had been lost at that time. However, since the environmental conditions for a more permanent colonization and establishment of *L. reuteri* Prodentis in the oral cavity had not been well determined [17], further studies are needed to confirm such a report.

Regarding the microbial profile associated with tooth 46, some interesting findings were observed after one month of probiotic consumption. The marked reduction in red complex species and the higher proportions of orange complex species could suggest a transition to simpler microbiota, with a predominance of more resistant species in the subgingival plaque, as assessed at T2 evaluation. The microbial profile identified for tooth 46 at the last follow-up assessment (T2) was consistent with the microbial succession described by Socransky et al. [5], which occurred mainly between the red and orange complexes during the transition from periodontal disease to periodontal health.

According to the data presented, in which an attenuation of fistulation that had presented in tooth 46 was observed, in addition to the establishment of a microbial profile with increased proportions of orange complex bacteria in the subgingival plaque of the whole mouth at the last follow-up evaluation, the use of the probiotic *L. reuteri* as monotherapy appears to be a viable and affordable option for the management of periodontal diseases. Therefore, the present case report represents an approach toward future perspectives, including case-control investigations using the probiotic *L. reuteri*, with and without periodontal mechanical therapy, and the use of new genomic technologies such as 16S gene analysis or culturomics to identify a larger number of subgingival strains [29].

The preservation of commensal microorganisms is essential for oral health as a fundamental part of overall health-related quality of life [1,30]. The use of probiotics as monotherapy for the treatment of periodontal diseases can be considered a promising adjunct for disabled or hospitalized individuals who are unable to perform usual oral care. Especially now, with the emergence of the COVID-19 pandemic [31], the data show that periodontitis is associated with an increased risk of intensive care unit (ICU) admission, the need for assisted ventilation, death in COVID-19 patients, and worse overall disease outcomes [32].

## 5. Conclusions

In the present study, it was observed that under monotherapy with *L. reuteri* Prodentis, the periodontal clinical measurements of the patient were maintained, while the subgingival microbiota presented selective changes, with increases of beneficial bacterial strains occurring over the main pathogenic subgingival microbiota.

The transient efficacy of the probiotic evaluated was directly proportional to the time of administration. However, in the most badly affected tooth evaluated (tooth number 46), the transition of the subgingival microbiota was maintained over the long term, even though the patient did not receive any additional periodontal treatment.

The results observed in the present study suggest a temporary clinical and selective antimicrobial effect of the probiotic *L. reuteri*, with efficacy toward the red complex species, and increased levels of beneficial *Actinomyces*, *Streptococcus*, *Gemella*, *Capnocytophaga*, and purple complex species during probiotic consumption for one month, suggesting that the probiotic *L. reuteri* Prodentis can be used as monotherapy for the control of periodontal diseases. However, further studies are required.

## Figures and Tables

**Figure 1 ijerph-19-07835-f001:**
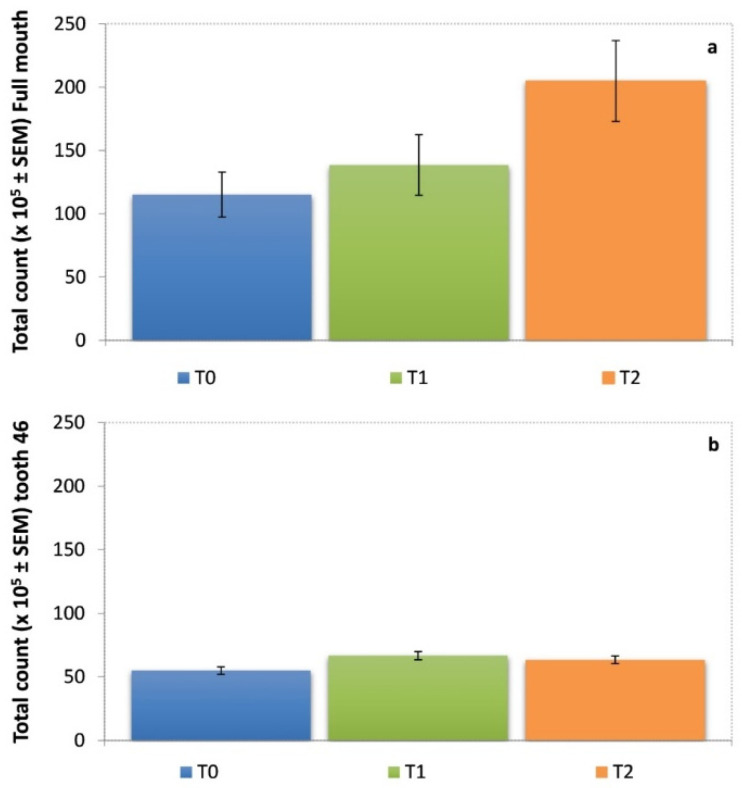
Mean total levels (total DNA probe count × 10^5^ ± SEM: Standard error of the mean) of 40 individual bacterial species: (**a**) whole-mouth subgingival plaque samples; and (**b**) subgingival plaque sample from tooth 46; at T0: baseline, T1: assessment at one month, and T2: assessment at three months.

**Figure 2 ijerph-19-07835-f002:**
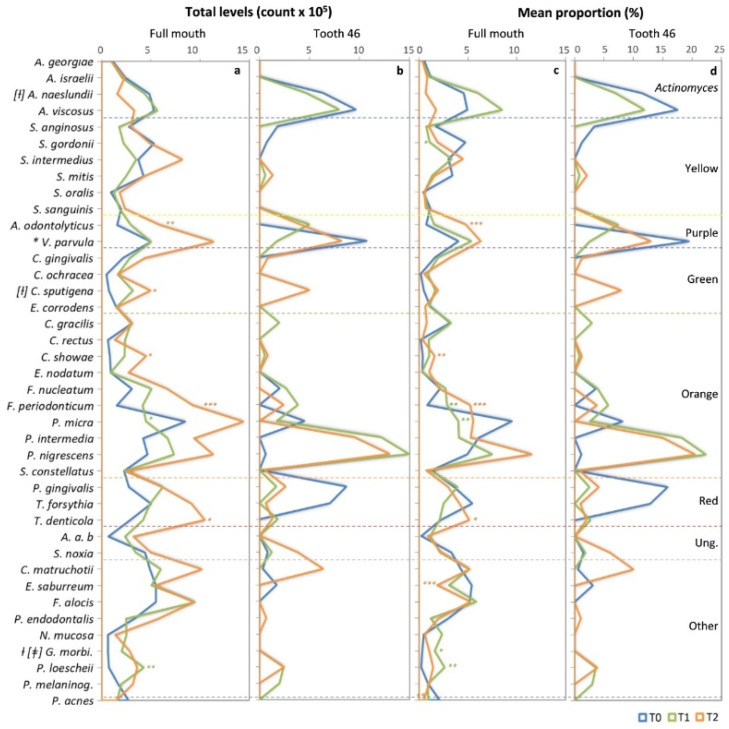
Mean total individual levels (total DNA probe count × 10^5^) and mean proportion (% total DNA probe count) of 40 individual bacterial species detected from the (**a**,**c**) full mouth subgingival plaque samples; (**b**,**d**) subgingival plaque sample from tooth 46; at T0: baseline, T1: assessment at one month, and T2: assessment at three months. The test species were grouped according to color complex and other species as similarly as possible to the description of microbial complexes in subgingival plaque by Socransky et al. [5]. “*A. a. b*.”: *Aggregatibacter actinomycetemcomitans* stp. b; “*G. morbi*.”: *Gemella morbillorum*; “*P. melaninog*.”: *P. melaninogenica*. Paired differences were determined using Kruskal–Wallis (in labels) for the counts: * *p <* 0.05, and † *p <* 0.01, proportions [†] *p <* 0.01, and [‡] *p <* 0.001, and Mann–Whitney U test, * *p <* 0.05, ** *p <* 0.01, and *** *p <* 0.001 Baseline vs. T2, of full mouth after adjustment for multiple comparisons.

**Figure 3 ijerph-19-07835-f003:**
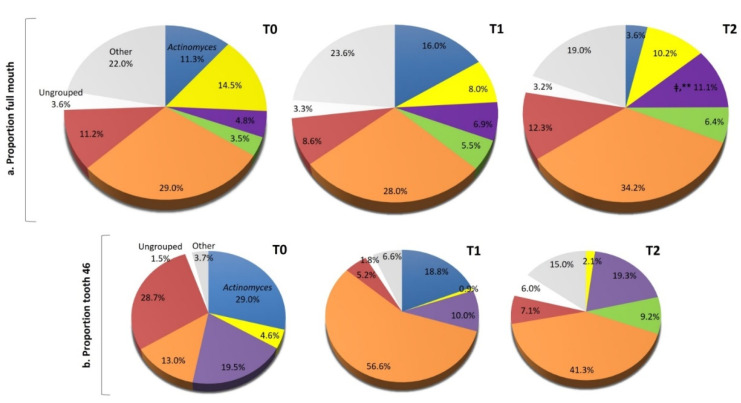
Pie charts of the mean proportions (% of total DNA probe count) of microbial complexes from (**a**) full mouth subgingival plaque samples; (**b**) subgingival plaque samples from tooth 46; at T0: baseline, T1: assessment at one month, and T2: assessment at three months. Taxa were grouped according to color complex according to the descriptions of microbial complexes in subgingival plaque [5]. Paired differences were determined using Kruskal–Wallis, ‡ *p <* 0.001, and Mann–Whitney U tests: ** *p <* 0.01 of full mouth after adjustment for multiple comparisons.

**Table 1 ijerph-19-07835-t001:** Reference strains employed for the development of DNA probes.

Species	ATCC	Complex	Species	ATCC	Complex
*Actinomyces georgiae*	49285	*Actinomyces*	*Neisseria mucosa*	19696	Other
*Actinomyces israelii*	12102	*Actinomyces*	*Parvimonas micra*	33270	Orange
*Actinomyces naeslundii*	12104	*Actinomyces*	*Porphyromonas endodontalis*	35406	Other
*Actinomyces odontolyticus*	17929	Purple	*Porphyromonas gingivalis*	33277	Red
*Actinomyces viscosus*	43146	*Actinomyces*	*Prevotella intermedia*	25611	Orange
*Aggregatibacter actinomycetemcomitans* stp. b.	43718	Ungrouped	*Prevotella melaninogenica*	25845	Other
*Campylobacter gracilis*	33236	Orange	*Prevotella nigrescens*	33563	Orange
*Campylobacter rectus*	33238	Orange	*Propionibacterium acnes*	6919	Other
*Campylobacter showae*	51146	Orange	*Selenomonas artemidis*	43528	Other
*Capnocytophaga gingivalis*	33624	Green	*Selenomonas noxia*	43541	Ungrouped
*Capnocytophaga ochracea*	27872	Green	*Streptococcus anginosus*	33397	Yellow
*Capnocytophaga sputigena*	33612	Green	*Streptococcus constellatus*	27823	Orange
*Corynebacterium matruchotii*	14266	Other	*Streptococcus gordonii*	10558	Yellow
*Eikenella corrodens*	23834	Green	*Streptococcus intermedius*	27335	Yellow
*Eubacterium saburreum*	33271	Other	*Streptococcus mitis*	49456	Yellow
*Eubacterium sulci*	35585	Other	*Streptococcus oralis*	35037	Yellow
*Fusobacterium nucleatum* subsp. *nucleatum*	25586	Orange	*Streptococcus sanguinis*	10556	Yellow
*Fusobacterium periodonticum*	33693	Orange	*Tannerella forsythia*	43037	Red
*Gemella morbillorum*	27824	Other	*Treponema denticola*	35405	Red
*Leptotrichia buccalis*	14201	Other	*Veillonella parvula*	10790	Purple

ATCC: American Type Culture Collection, Rockville, MD, USA. Complex: Species were grouped according to color, similar to descriptions of microbial complexes in subgingival plaque [5], with the following exceptions: *C.*
*matruchotii*, *E.*
*saburreum*, *E.*
*sulci*, *G.*
*morbillorum*, *L.*
*buccalis*, *N.*
*mucosa*, and *P.*
*endodontalis*, *P.*
*melaninogenica*, *P.*
*acnes*, and *S.*
*artemidis* were grouped as “Other”. Stp.: serotype. Subsp.: subspecies.

**Table 2 ijerph-19-07835-t002:** Clinical periodontal characteristics of the patient in full mouth and tooth number 46.

Full Mouth	T0		T1		T2	
	**Media**	**±SEM**	**Media**	**±SEM**	**Media**	**±SEM**
Mean pocket depth (millimeters, mm)	3.9	0.2	3.3	0.2	3.4	0.2
Mean attachment level (AL, mm)	2	0.2	1.6	0.1	2.0	0.2
Mean number sites with AL ≥ 5 mm	8	1.1	4	1.5	6	1.4
% sites with:						
Plaque accumulation (percent, %)	14.5	0.7	8.7	1.1	14.5	0.7
**^,^ ‡ Gingival erythema (%)	45.7	0.1	44.2	0.1	69.6	0.1
* Bleeding on probing (%)	74.6	0.1	58	0.1	98.6	0
Suppuration (%)	17.4	0.1	25.4	0.1	24.6	0.1
**Tooth 46**	**T0**		**T1**		**T2**	
	**Media**	**±SEM**	**Media**	**±SEM**	**Media**	**±SEM**
Mean pocket depth (mm)	10.2	1.6	8.9	1.8	8.8	1.7
Mean attachment level (AL, mm)	7.4	1.5	6.3	1.5	6.7	1.6
Mean number sites with AL ≥ 5 mm	4	1.2	3	0.3	3	0.6
% sites with:						
Plaque accumulation (%)	33.3	0.2	16.7	0.2	50	0.2
∞, † Gingival erythema (%)	100	0.0	16.7	0.2	100	0
Bleeding on probing (%)	33.3	0.2	33.3	0.2	66.7	0.2
Suppuration (%)	0.5	0.2	0	0	0	0

Paired differences wPaired differences were determined using Kruskalkal–Wallis: † *p* < 0.01, and ‡ *p* < 0.001, and Mann–Whitney U tests, * *p* < 0.05, ** *p* < 0.01 (Baseline vs. T2, full mouth), and ∞ *p* < 0.05 (Baseline vs. T1, tooth 46) after adjustment for multiple comparisons. SEM: Standard error of the mean.

## Data Availability

The authors declare that they have followed the protocols of the School of Dentistry of the National Autonomous University of Mexico (UNAM) on the publication of patient data. The corresponding author is the owner of this document, as well as the data results when requested.

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
