# Peer review of "Probiotic Monotherapy with Lactobacillus reuteri (Prodentis) as a Coadjutant to Reduce Subgingival Dysbiosis in a Patient with Periodontitis"

_ijerph, 2022, doi:10.3390/ijerph19137835_

Round 1

Reviewer 1 Report

Dear authors,

thanks for your contribution. Even if it is very interesting some major concerns must be fixed before going ahead in its editorial process.

In details:

  1. English language revision is strongly needed, please let the manuscript be revised by a native speaker;
  2. A high number of references is too old-fashioned and must be updated with new techniques (see comments below for more infos);
  3. All the commodity data of the product (Prodentis) must be clearly reported;
  4. The choice of analyzing through an ibridazione technique only a limited number of bacteria is a major limit of the paper, why this choice was made?
  5. Given the fact that the patient had several missing elements (even in aesthetic sectors: upper central and lateral incisors) did she wear a removable prosthesis? Please make this element clear as it can interfere and modify the oral microbiota.
  6. The first lower left molar was deeply decayed, this could have interfered with the microbiota composition, was this considered?
  7. Please dedicate a paragraph to new cultural and genomic methods for the microbiota characterization: "Characterizing peri-implant and sub-gingival microbiota through culturomics. First isolation of some species in the oral cavity. A pilot study", DOI: 10.3390/pathogens9050365.

Author Response

The coauthors and I gratefully acknowledge the valuable comments of the reviewers. Changes made to the manuscript before resubmission for editorial evaluation are detailed below. The changes are tracked with the corresponding word tool in the paper:

Manuscript ID: ijerph-1733684

Type of manuscript: Case Report

Title: Probiotic monotherapy with L. reuteri Prodentis as a coadjutant to

reduce subgingival dysbiosis in a patient with periodontitis.

Authors: Claudia Salinas-Azuceno, Miryam Martínez-Hernández, José-Isaac

Maldonado-Noriega, Adriana-Patricia Rodríguez-Hernández *, Laurie-Ann

Ximenez-Fyvie

Response to Reviewer 1 Comments

Point 1: English language revision is strongly needed, please let the manuscript be revised by a native speaker;

Response 1: The manuscript was revised and corrected by MDPI English Editing service.

Point 2: A high number of references is too old-fashioned and must be updated with new techniques (see comments below for more infos);

Response 2: The following references, which appeared in the first version of the manuscript as follows, were either deleted or updated:

  1. Southerland, J. H.; Taylor, G. W.; Moss, K.; Beck, J. D.; Offenbacher, S., Commonality in chronic inflammatory diseases: periodontitis, diabetes, and coronary artery disease. Periodontology 2000 2006, 40, (1), 130-143.
  2. Kassebaum, N.; Bernabé, E.; Dahiya, M.; Bhandari, B.; Murray, C.; Marcenes, W., Global burden of severe periodontitis in 1990-2010: a systematic review and meta-regression. Journal of dental research 2014, 93, (11), 1045-1053.
  3. Cobb, C. M., Non‐surgical pocket therapy: Mechanical. Annals of periodontology 1996, 1, (1), 443-490.
  4. Fooks, L.; Gibson, G. R., Probiotics as modulators of the gut flora. British Journal of Nutrition 2002, 88, (S1), s39-s49.
  5. Arora, T.; Singh, S.; Sharma, R.K., Probiotics: interaction with gut microbiome and antiobesity potential. Nutrition 2013, 29, (4), 591-596.
  6. Näse, L.; Hatakka, K.; Savilahti, E.; Saxelin, M.; Pönkä, A.; Poussa, T.; Korpela, R.; Meurman, J. H., Effect of long–term con-sumption of a probiotic bacterium, Lactobacillus rhamnosus GG, in milk on dental caries and caries risk in children. Caries research 2001, 35, (6), 412-420.
  7. Haffajee, A. D.; Socransky, S. S.; Goodson, J. M., Clinical parameters as predictors of destructive periodontal disease activ-ity. Journal of clinical periodontology 1983, 10, (3), 257-65.
  8. Feinberg, A. P.; Vogelstein, B., A technique for radiolabeling DNA restriction endonuclease fragments to high specific activity. Anal Biochem 1983, 132, (1), 6-13.
  9. Socransky, S. S.; Smith, C.; Martin, L.; Paster, B. J.; Dewhirst, F. E.; Levin, A. E., "Checkerboard" DNA-DNA hybridization. Biotechniques 1994, 17, (4), 788-92.
  10. Smith, G. L.; Socransky, S. S.; Sansone, C., "Reverse" DNA hybridization method for the rapid identification of subgingival microorganisms. Oral Microbiol Immunol 1989, 4, (3), 141-5.
  11. Socransky, S. S.; Haffajee, A. D.; Cugini, M. A.; Smith, C.; Kent, R. L., Jr., Microbial complexes in subgingival plaque. Jour-nal of clinical periodontology 1998, 25, (2), 134-44.
  12. Socransky, S. S.; Haffajee, A. D.; Smith, C.; Dibart, S., Relation of counts of microbial species to clinical status at the sam-pled site. Journal of clinical periodontology 1991, 18, (10), 766-75.
  13. Jacobsen, C. N.; Nielsen, V. R.; Hayford, A.; Møller, P.; Michaelsen, K.; Paerregaard, A.; Sandström, B.; Tvede, M.; Jakobsen, M., Screening of probiotic activities of forty-seven strains of Lactobacillus spp. by in vitro techniques and evaluation of the colonization ability of five selected strains in humans. Applied and environmental microbiology 1999, 65, (11), 4949-4956.
  14. Shimauchi, H.; Mayanagi, G.; Nakaya, S.; Minamibuchi, M.; Ito, Y.; Yamaki, K.; Hirata, H., Improvement of periodontal condition by probiotics with Lactobacillus salivarius WB21: a randomized, double‐blind, placebo‐controlled study. Journal of clinical periodontology 2008, 35, (10), 897-905.
  15. Twetman, S.; Derawi, B.; Keller, M.; Ekstrand, K.; Yucel-Lindberg, T.; Stecksen-Blicks, C., Short-term effect of chewing gums containing probiotic Lactobacillus reuteri on the levels of inflammatory mediators in gingival crevicular fluid. Acta odontologica Scandinavica 2009, 67, (1), 19-24.
  16. Herrera, D.; Alonso, B.; de Arriba, L.; Santa Cruz, I.; Serrano, C.; Sanz, M., Acute periodontal lesions. Periodontology 2000 2014, 65, (1), 149-177.
  17. Gift, H. C.; Atchison, K. A., Oral health, health, and health-related quality of life. Medical care 1995, NS57-NS77.
  18. Naito, M.; Yuasa, H.; Nomura, Y.; Nakayama, T.; Hamajima, N.; Hanada, N., Oral health status and health-related quality of life: a systematic review. Journal of oral science 2006, 48, (1), 1-7.

In addition, we added more recent references to update the bibliography of the manuscript:

  1. Genco, R. J.; Sanz, M., Clinical and public health implications of periodontal and systemic diseases: An overview. Periodontology 2000 2020, 83, (1), 7-13.
  2. Martínez-García, M.; Hernández-Lemus, E., Periodontal inflammation and systemic diseases: an overview. Frontiers in physiology 2021, 1842.
  3. Feng, P.; Ye, Z.; Kakade, A.; Virk, A. K.; Li, X.; Liu, P., A review on gut remediation of selected environmental contaminants: possible roles of probiotics and gut microbiota. Nutrients 2018, 11, (1), 22.
  4. Ahuja, A.; Ahuja, V., Efficacy of probiotic Lactobacillus reuteri on chronic generalised periodontitis: A systematic review of randomized controlled clinical trials in humans. The Journal of Dental Panacea 2021, 3, (3), 106-117.
  5. Sinkiewicz, G.; Cronholm, S.; Ljunggren, L.; Dahlen, G.; Bratthall, G., Influence of dietary supplementation with Lactobacillus reuteri on the oral flora of healthy subjects. Swedish dental journal 2010, 34, (4), 197-206.
  6. Hallström, H.; Lindgren, S.; Yucel-Lindberg, T.; Dahlén, G.; Renvert, S.; Twetman, S., Effect of probiotic lozenges on inflammatory reactions and oral biofilm during experimental gingivitis. Acta odontologica Scandinavica 2013, 71, (3-4), 828-833.
  7. Rodríguez-Hernández, A.-P.; Márquez-Corona, M. d. L.; Pontigo-Loyola, A. P.; Medina-Solís, C. E.; Ximenez-Fyvie, L.-A., Subgingival microbiota of Mexicans with type 2 diabetes with different periodontal and metabolic conditions. International journal of environmental research and public health 2019, 16, (17), 3184.
  8. Socransky, S.; Haffajee, A.; Smith, C.; Martin, L.; Haffajee, J.; Uzel, N.; Goodson, J., Use of checkerboard DNA–DNA hybridization to study complex microbial ecosystems. Oral microbiology and immunology 2004, 19, (6), 352-362.
  9. Teles, F. R., The microbiome of peri-implantitis: is it unique. Compend. Contin. Educ. Dent 2017, 38, (8), 22-25.
  10. Martellacci, L.; Quaranta, G.; Patini, R.; Isola, G.; Gallenzi, P.; Masucci, L., A literature review of metagenomics and culturomics of the peri-implant microbiome: Current evidence and future perspectives. Materials 2019, 12, (18), 3010.
  11. Schierz, O.; Baba, K.; Fueki, K., Functional oral health‐related quality of life impact: A systematic review in populations with tooth loss. Journal of Oral Rehabilitation 2021, 48, (3), 256-270.

Point 3: All the commodity data of the product (Prodentis) must be clearly reported;

Response 3:

We agree with the reviewer, the specifications were added, and now appear on page: 3, lines: 118 – 120:

“… fully hydrogenated palm oil, peppermint flavor, menthol flavor, peppermint oil, and sucralose. Each lozenge was composed of a minimum of 200 million live L. reuteri Prodentis.”

Point 4: The choice of analyzing through an ibridazione technique only a limited number of bacteria is a major limit of the paper, why this choice was made?

Response 4:

We decided to identify changes in the levels and proportions of the subgingival microbiota using checkerboard DNA–DNA hybridization technique, as it is a useful tool for the enumeration of bacterial species in microbiologically complex systems and offers the possibility to quantify representative strains strongly related to the pathogenesis of commensal species such as periodontitis or peri-implantitis. Recently Teles, F. et all. reported that the checkerboard DNA-DNA hybridization technique quickly became the leading method for studies of peri-implant microbiota [Socransky, S.; et al. Oral microbiology and immunology 2004, 19, (6), 352-362.]. The technique provides a cost-effective platform for the detection and enumeration of the 40 most common periodontal bacterial species in multiple samples simultaneously [Teles, F. R. Compend. Contin. Educ. Dent 2017, 38, (8), 22-25.]. Despite the advantages that checkerboard studies guarantee, it is important to note that the technique is limited to the detection of the taxa targeted by the probes used, which represents a limitation compared to high-throughput next-generation sequencing (NGS) approaches that can detect a large number of species. Furthermore, although 16S rRNA provides a broad view of sample composition, it may not distinguish closely related species, in addition to the fact that this technique ignores most of the microbial genetic information contained in the sample because it focuses on a portion of a bacterial gene [Teles, F. R. Compend. Contin. Educ. Dent 2017, 38, (8), 22-25. Wu, R.; et al. Journal of dental research 2011, 90, (5), 561-572].

To clarify this point in the manuscript, we add the following paragraph on the page 10, lines 363 – 368:

“It should be noted that an advantage of using the Checkerboard technique, as in the pre-sent study, is the possibility for enumerating 40 bacterial species in microbiologically complex systems, and provision of the quantification of representative strains that are strongly related to the pathogenesis of periodontitis is crucial for a periodontal microbial diagnosis [27, 28].”

Socransky, S.; Haffajee, A.; Smith, C.; Martin, L.; Haffajee, J.; Uzel, N.; Goodson, J., Use of checkerboard DNA–DNA hybridization to study complex microbial ecosystems. Oral microbiology and immunology 2004, 19, (6), 352-362.

Teles, F. R., The microbiome of peri-implantitis: is it unique?. Compend. Contin. Educ. Dent 2017, 38, (8), 22-25.

Wu, R.; Zhao, X.; Wang, Z.; Zhou, M.; Chen, Q., Novel molecular events in oral carcinogenesis via integrative approaches. Journal of dental research 2011, 90, (5), 561-572.

Point 5.           Given the fact that the patient had several missing elements (even in aesthetic sectors: upper central and lateral incisors) did she wear a removable prosthesis? Please make this element clear as it can interfere and modify the oral microbiota.

Response 5: The patient was wearing a removable acrylic prosthesis in the upper central and lateral sector, however, it is important to note that according to the literature the microbiota associated with removable prostheses is significantly different in both composition and diversity from the subgingival biofilm and oral microbiome composition [O'Donnell, et al. Plos one 2015, 10, (9), e0137717. Marchi-Alves, L. M.; et al. BioMed Research International 2017, 2017]. Furthermore, the removable prosthesis has no influence on the subgingival microbiota, as it is not in direct contact with the surrounding teeth.

Point 6.           The first lower left molar was deeply decayed, this could have interfered with the microbiota composition, was this considered?

Response 6: The lower left first molar was deeply carious, however, there is no conclusive evidence that the caries microbiota can influence the subgingival microbiota, as caries and subgingival biofilm are site-specific. In fact, DNA sequencing studies of these different habitats have revealed that both habitats harbored different microbial communities [Valm, A. M. Journal of molecular biology 2019, 431, (16), 2957-2969].

Point 7.           Please dedicate a paragraph to new cultural and genomic methods for the microbiota characterization: "Characterizing peri-implant and sub-gingival microbiota through culturomics. First isolation of some species in the oral cavity. A pilot study", DOI: 10.3390/pathogens9050365.

Response 7: We are aware that newer technologies exist to identify a greater number of oral bacteria, so in the text, we add information relevant to sequencing and culturomics approaches which are an excellent option to detect a greater number of subgingival strains, page 12, lines 417 - 421:

“Therefore, the present case report represents an approach toward future perspectives, including case-control investigations using the probiotic L. reuteri, with and without periodontal mechanical therapy, and the use of new genomic technologies such as 16S gene analysis or culturomics to identify a larger number of subgingival strains [29].”

Reviewer 2 Report

1. In the introduction, further description of the therapeutic effect and role of L. reuteri Prodentis is needed.

2. An explanation of beneficial bacteria and bacteria in the result is necessary in the introduction section.

3. Criteria for selecting tooth #46 and explanation required.

4. In Table 2, T0->T1 decreased, but it appeared to increase in T0->T2. Further consideration is needed on the difference between the results in Table 2 and the duration of administration.

5. It is necessary to describe the limitations of the study for a study involving a single institution.

6. In this study, mechanical periodontal teratment was not performed, so it is not appropriate to compare it with the study performed (line 336-341).

7. It is necessary to consider the effectiveness of L. reuteri on #46 (apical inflammation) teeth compared to full mouth (periodontal disease target teeth)

Author Response

The coauthors and I gratefully acknowledge the valuable comments of the reviewers. Changes made to the manuscript before resubmission for editorial evaluation are detailed below. The changes are tracked with the corresponding word tool in the paper:

Manuscript ID: ijerph-1733684

Type of manuscript: Case Report

Title: Probiotic monotherapy with L. reuteri Prodentis as a coadjutant to

reduce subgingival dysbiosis in a patient with periodontitis.

Authors: Claudia Salinas-Azuceno, Miryam Martínez-Hernández, José-Isaac

Maldonado-Noriega, Adriana-Patricia Rodríguez-Hernández *, Laurie-Ann

Ximenez-Fyvie

Response to Reviewer 2 Comments

Point 1. In the introduction, further description of the therapeutic effect and role of L. reuteri Prodentis is needed.

Response 1: To respond to this comment:

“Among them, the probiotic L. reuteri is beneficial in improving the clinical and microbiological parameters that are associated with periodontitis, by resolving inflammation and by reducing the molecular mediators associated with bone loss [11-14]. The probiotic L. reuteri has been recognized as a homeostasis enhancer by promoting beneficial subgingival strains on major periodontal pathogens [15, 16]. In addition, and given that contradictory results refute the inflammatory or microbiological benefits of L. reuteri [17, 18], the present case report represents a research gap supporting monotherapy for the management of periodontitis, ….” On page 2, lines 69-77 was rewritten.

Point 2. An explanation of beneficial bacteria and bacteria in the result is necessary in the introduction section.

Response 2: By the reviewer's observation, we add in the introduction information relevant to the interactions between the main pathogens and beneficial bacteria in the process of dysbiosis. Page 1, lines 41 - 46:

“Periodontitis is an infectious and inflammatory disorder that is characterized by the destruction of the supporting structure of teeth, induced by biofilm dysbiosis, in which a higher proportion of major pathogens such as red complex species, including Porphyromonas gingivalis, Tannerella forsythia, and Treponema denticola, and a low proportion of beneficial strains, such as Actinomyces sp. or Streptococcus sp., remain the key to microbial disruption [3-5].”

Point 3. Criteria for selecting tooth #46 and explanation required.

Response 3: A detailed analysis of tooth 46 was done since at baseline it presented with an active periradicular process, showing significant clinical improvement at the end of follow-up (T2). On page 5, lines 193 - 196:

“Special attention was paid to the right mandibular first molar (numbered 46, according to FDI nomenclature), in which an active periradicular infection with vestibular fistulation and radiography bone loss (4 sites with AL≥5mm, T0) due to an endodontic failure was observed during the baseline evaluation;...”

Point 4. In Table 2, T0->T1 decreased, but it appeared to increase in T0->T2. Further consideration is needed on the difference between the results in Table 2 and the duration of administration.

Response 4: This information is specified in the Results section on Page 6, lines 212 -214:

“Additionally, PLA (T2: 69.6 %, p<0.01 UM, p<0.05 KW), GE (T2: 98.6 %, p<0.01 UM), and BOP (T2: 66.7 %, p<0.05 MW) were clinical parameters that showed a significantly higher value at T2, compared to T0 evaluation (Table 2).”

And in the conclusions section, on Page 11, lines 447 – 451, it is written the following:

“The transient efficacy of the probiotic evaluated was directly proportional to the time of administration. However, in the most badly affected tooth evaluated (tooth number 46), the transition of the subgingival microbiota was maintained over the long term, even though the patient did not receive any additional periodontal treatment.”

Point 5. It is necessary to describe the limitations of the study for a study involving a single institution. Revisor: The choice of analyzing through an ibridazione technique only a limited number of bacteria is a major limit of the paper, why this choice was made?

Response 5: This is a case report and is therefore subject to certain limitations. However, by using the checkerboard technique, we were able to obtain much information regarding the changes in the subgingival microbiota resulting from the application of L. reutei as monotherapy. To clarify this point, we specify the advantages of the technique employed on page 11, lines 363 - 368:

“It should be noted that an advantage of using the Checkerboard technique, as in the pre-sent study, is the possibility for enumerating 40 bacterial species in microbiologically complex systems, and provision of the quantification of representative strains that are strongly related to the pathogenesis of periodontitis is crucial for a periodontal microbial diagnosis [27, 28].”

and then, on the page 11 lines 417 - 421:

“Therefore, the present case report represents an approach toward future perspectives, including case-control investigations using the probiotic L. reuteri, with and without periodontal mechanical therapy, and the use of new genomic technologies such as 16S gene analysis or culturomics to identify a larger number of subgingival strains [29].”

Point 6. In this study, mechanical periodontal teratment was not performed, so it is not appropriate to compare it with the study performed (line 336-341).

Response 6: As it was not aimed at comparing mechanotherapy with the use of Lactobacillus reuteri probiotic but highlight the improvement of clinical parameters related to the fistulation in teeth 46 after the use of probiotics as monotherapy, the paragraph was rewritten as follows.

“In this case report, a significant reduction in gingival erythema was observed in tooth 46 at T1 evaluation, and in addition to a slight reduction in pocket depth, AL, and plaque accumulation, these findings were associated with fistula regression in tooth 46 when the probiotic L. reuteri was employed as monotherapy." (On Pages 10 and 11 lines 354 - 359).

Point 7. It is necessary to consider the effectiveness of L. reuteri on #46 (apical inflammation) teeth compared to full mouth (periodontal disease target teeth).

Response 7: Table 2 presents information relevant to the efficacy of L. reuteri monotherapy in attenuating the periradicular lesion (#46) compared to full-mouth clinical parameters. Unfortunately, in the present clinical case, we didn't take apical samples since it would require major surgery.

Reviewer 3 Report

Thank you for allowing me to review this manuscript.

This paper deals with a topic of great interest in the odontostomatology field. Still, I believe that the authors should highlight the limitations of this study, considering that it involves only one patient.

I also suggest that the English language, style, and grammar be checked

Author Response

The coauthors and I gratefully acknowledge the valuable comments of the reviewers. Changes made to the manuscript before resubmission for editorial evaluation are detailed below. The changes are tracked with the corresponding word tool in the paper:

Manuscript ID: ijerph-1733684

Type of manuscript: Case Report

Title: Probiotic monotherapy with L. reuteri Prodentis as a coadjutant to

reduce subgingival dysbiosis in a patient with periodontitis.

Authors: Claudia Salinas-Azuceno, Miryam Martínez-Hernández, José-Isaac

Maldonado-Noriega, Adriana-Patricia Rodríguez-Hernández *, Laurie-Ann

Ximenez-Fyvie

Response to Reviewer 3 Comments

Point 1. This paper deals with a topic of great interest in the odontostomatology field. Still, I believe that the authors should highlight the limitations of this study, considering that it involves only one patient.

Response 1: This is a case report and is therefore subject to certain limitations, however, the following information was added on page 11 lines 417 - 421:

“Therefore, the present case report represents an approach toward future perspectives, including case-control investigations using the probiotic L. reuteri, with and without periodontal mechanical therapy, and the use of new genomic technologies such as 16S gene analysis or culturomics to identify a larger number of subgingival strains [29].” to note the limitation and future perspectives.

Point 2. I also suggest that the English language, style, and grammar be checked

Response 2: The manuscript was revised and corrected by MDPI English Editing service.

Reviewer 4 Report

Dear Authors ,

The present case report is an attempt to assess Probiotic monotherapy with L. reuteri Prodentis as a coadjutant  to reduce subgingival dysbiosis in a patient with periodontitis. As per the systematic review and meta analysis by Gheisary et al. 2022 ,the Probiotic therapy as an adjuct to mechanical therapy in treatment of periodontal disease has shown benefits . However, its usage as a mono therapy needs further investigation as the studies are scanty . This case report will help the scientific community

Introduction :

1.      The authors have to elaborate more on the oral dysbiosis mechanism as the aim of the study is to see the effect of subgingival dysbiosis with the application of L. reuteri Prodentis . I would suggest the author should take a reference of  https://doi.org/10.3390/microorganisms9091966

2.      The authors have cited many articles which as dated before 2012 . I would suggest the authors to update the references

3.      The authors in the introductions should mention the previous study which was conducted as a monotherapy as well as the present status of the probiotic therapy in the periodontal disease . The authors can take a reference of  https://doi.org/10.3390/nu14051036

4.      The authors should elaborate on the etiopathogenesis of the periodontal disease and an emphasis on systemic inflammation . the authors can take a of reference https://doi.org/10.1038/s41577-020-00488-6 , https://doi.org/10.3389/fphys.2021.709438

5.      The author should mention the research gap in the introduction . The authors have mentioned few studies in the discussion section . I would suggest this should be written in introduction section and state the novelty of the current study .

Material and method :

1.      Kindly mention  the stage and grade of periodontitis patient

Results :

1.      Clinical / microbiological pictures can be incorporated 

Discussion :

1.      Kindly comment why the authors have chosen the study as a case report . Why they have not recruited more study samples to complete it as a original article or case series ?

2.      Write limitations and future directions as separate headings

Regards

Author Response

The coauthors and I gratefully acknowledge the valuable comments of the reviewers. Changes made to the manuscript before resubmission for editorial evaluation are detailed below. The changes are tracked with the corresponding word tool in the paper:

Manuscript ID: ijerph-1733684

Type of manuscript: Case Report

Title: Probiotic monotherapy with L. reuteri Prodentis as a coadjutant to

reduce subgingival dysbiosis in a patient with periodontitis.

Authors: Claudia Salinas-Azuceno, Miryam Martínez-Hernández, José-Isaac

Maldonado-Noriega, Adriana-Patricia Rodríguez-Hernández *, Laurie-Ann

Ximenez-Fyvie

Response to Reviewer 4 Comments

Introduction:

Point 1. The authors have to elaborate more on the oral dysbiosis mechanism as the aim of the study is to see the effect of subgingival dysbiosis with the application of L. reuteri Prodentis. I would suggest the author should take a reference of https://doi.org/10.3390/microorganisms9091966

Response 1: By the reviewer's comment, the following information was added on pages 1 and 2, lines 34 - 47:

“The oral cavity is the window of the body, and it is home to a diverse microbial community encompassing one of the most complex microbiomes and dynamic microbial communities, comprising hundreds of different species of bacteria living in a structure known as a biofilm. Dysbiosis of the periodontal microbiota can interfere with the normal function of the host immune system, resulting in the development of inflammatory dis-eases, such as periodontal disease [1-3]. Periodontitis is an infectious and inflammatory disorder that is characterized by the destruction of the supporting structure of teeth, induced by biofilm dysbiosis, in which a higher proportion of major pathogens such as red complex species, including Porphyromonas gingivalis, Tannerella forsythia, and Treponema denticola, and a low proportion of beneficial strains, such as Actinomyces sp. or Streptococcus sp., remain the key to microbial disruption [3-5].”

Point 2. The authors have cited many articles which as dated before 2012. I would suggest the authors to update the references.

Response 2: In agreement with the reviewer´s comment, the following references, which appeared in the first version of the manuscript as follows, were either deleted or updated:

5. Southerland, J. H.; Taylor, G. W.; Moss, K.; Beck, J. D.; Offenbacher, S., Commonality in chronic inflammatory diseases: periodontitis, diabetes, and coronary artery disease. Periodontology 2000 2006, 40, (1), 130-143.

6. Kassebaum, N.; Bernabé, E.; Dahiya, M.; Bhandari, B.; Murray, C.; Marcenes, W., Global burden of severe periodontitis in 1990-2010: a systematic review and meta-regression. Journal of dental research 2014, 93, (11), 1045-1053.

8. Cobb, C. M., Non‐surgical pocket therapy: Mechanical. Annals of periodontology 1996, 1, (1), 443-490.

10. Fooks, L.; Gibson, G. R., Probiotics as modulators of the gut flora. British Journal of Nutrition 2002, 88, (S1), s39-s49.

11. Arora, T.; Singh, S.; Sharma, R.K., Probiotics: interaction with gut microbiome and antiobesity potential. Nutrition 2013, 29, (4), 591-596.

13. Näse, L.; Hatakka, K.; Savilahti, E.; Saxelin, M.; Pönkä, A.; Poussa, T.; Korpela, R.; Meurman, J. H., Effect of long–term con-sumption of a probiotic bacterium, Lactobacillus rhamnosus GG, in milk on dental caries and caries risk in children. Caries research 2001, 35, (6), 412-420.

18. Haffajee, A. D.; Socransky, S. S.; Goodson, J. M., Clinical parameters as predictors of destructive periodontal disease activ-ity. Journal of clinical periodontology 1983, 10, (3), 257-65.

20. Feinberg, A. P.; Vogelstein, B., A technique for radiolabeling DNA restriction endonuclease fragments to high specific activity. Anal Biochem 1983, 132, (1), 6-13.

21. Socransky, S. S.; Smith, C.; Martin, L.; Paster, B. J.; Dewhirst, F. E.; Levin, A. E., "Checkerboard" DNA-DNA hybridization. Biotechniques 1994, 17, (4), 788-92.

22. Smith, G. L.; Socransky, S. S.; Sansone, C., "Reverse" DNA hybridization method for the rapid identification of subgingival microorganisms. Oral Microbiol Immunol 1989, 4, (3), 141-5.

23. Socransky, S. S.; Haffajee, A. D.; Cugini, M. A.; Smith, C.; Kent, R. L., Jr., Microbial complexes in subgingival plaque. Jour-nal of clinical periodontology 1998, 25, (2), 134-44.

25. Socransky, S. S.; Haffajee, A. D.; Smith, C.; Dibart, S., Relation of counts of microbial species to clinical status at the sam-pled site. Journal of clinical periodontology 1991, 18, (10), 766-75.

29. Jacobsen, C. N.; Nielsen, V. R.; Hayford, A.; Møller, P.; Michaelsen, K.; Paerregaard, A.; Sandström, B.; Tvede, M.; Jakobsen, M., Screening of probiotic activities of forty-seven strains of Lactobacillus spp. by in vitro techniques and evaluation of the colonization ability of five selected strains in humans. Applied and environmental microbiology 1999, 65, (11), 4949-4956.

32. Shimauchi, H.; Mayanagi, G.; Nakaya, S.; Minamibuchi, M.; Ito, Y.; Yamaki, K.; Hirata, H., Improvement of periodontal condition by probiotics with Lactobacillus salivarius WB21: a randomized, double‐blind, placebo‐controlled study. Journal of clinical periodontology 2008, 35, (10), 897-905.

33. Twetman, S.; Derawi, B.; Keller, M.; Ekstrand, K.; Yucel-Lindberg, T.; Stecksen-Blicks, C., Short-term effect of chewing gums containing probiotic Lactobacillus reuteri on the levels of inflammatory mediators in gingival crevicular fluid. Acta odontologica Scandinavica 2009, 67, (1), 19-24.

34. Herrera, D.; Alonso, B.; de Arriba, L.; Santa Cruz, I.; Serrano, C.; Sanz, M., Acute periodontal lesions. Periodontology 2000 2014, 65, (1), 149-177.

39. Gift, H. C.; Atchison, K. A., Oral health, health, and health-related quality of life. Medical care 1995, NS57-NS77.

40. Naito, M.; Yuasa, H.; Nomura, Y.; Nakayama, T.; Hamajima, N.; Hanada, N., Oral health status and health-related quality of life: a systematic review. Journal of oral science 2006, 48, (1), 1-7.

In addition, we added more recent references to update the bibliography:

6. Genco, R. J.; Sanz, M., Clinical and public health implications of periodontal and systemic diseases: An overview. Periodontology 2000 2020, 83, (1), 7-13.

7. Martínez-García, M.; Hernández-Lemus, E., Periodontal inflammation and systemic diseases: an overview. Frontiers in physiology 2021, 1842.

10.Feng, P.; Ye, Z.; Kakade, A.; Virk, A. K.; Li, X.; Liu, P., A review on gut remediation of selected environmental contaminants: possible roles of probiotics and gut microbiota. Nutrients 2018, 11, (1), 22.

15. Ahuja, A.; Ahuja, V., Efficacy of probiotic Lactobacillus reuteri on chronic generalised periodontitis: A systematic review of randomized controlled clinical trials in humans. The Journal of Dental Panacea 2021, 3, (3), 106-117.

17. Sinkiewicz, G.; Cronholm, S.; Ljunggren, L.; Dahlen, G.; Bratthall, G., Influence of dietary supplementation with Lactobacillus reuteri on the oral flora of healthy subjects. Swedish dental journal 2010, 34, (4), 197-206.

18. Hallström, H.; Lindgren, S.; Yucel-Lindberg, T.; Dahlén, G.; Renvert, S.; Twetman, S., Effect of probiotic lozenges on inflammatory reactions and oral biofilm during experimental gingivitis. Acta odontologica Scandinavica 2013, 71, (3-4), 828-833.

19. Rodríguez-Hernández, A.-P.; Márquez-Corona, M. d. L.; Pontigo-Loyola, A. P.; Medina-Solís, C. E.; Ximenez-Fyvie, L.-A., Subgingival microbiota of Mexicans with type 2 diabetes with different periodontal and metabolic conditions. International journal of environmental research and public health 2019, 16, (17), 3184.

27. Socransky, S.; Haffajee, A.; Smith, C.; Martin, L.; Haffajee, J.; Uzel, N.; Goodson, J., Use of checkerboard DNA–DNA hybridization to study complex microbial ecosystems. Oral microbiology and immunology 2004, 19, (6), 352-362.

28. Teles, F. R., The microbiome of peri-implantitis: is it unique. Compend. Contin. Educ. Dent 2017, 38, (8), 22-25.

29. Martellacci, L.; Quaranta, G.; Patini, R.; Isola, G.; Gallenzi, P.; Masucci, L., A literature review of metagenomics and culturomics of the peri-implant microbiome: Current evidence and future perspectives. Materials 2019, 12, (18), 3010.

30.       Schierz, O.; Baba, K.; Fueki, K., Functional oral health‐related quality of life impact: A systematic review in populations with tooth loss. Journal of Oral Rehabilitation 2021, 48, (3), 256-270.

Point 3. The authors in the introductions should mention the previous study which was conducted as a monotherapy as well as the present status of the probiotic therapy in the periodontal disease. The authors can take a reference of https://doi.org/10.3390/nu14051036.

Response 3: Following the reviewers' comments, we added more information related to probiotic therapy in the Introduction section on page 2, lines 69 - 80.

“Among them, the probiotic L. reuteri is beneficial in improving the clinical and microbiological parameters that are associated with periodontitis, by resolving inflammation and by reducing the molecular mediators associated with bone loss [11-14]. The probiotic L. reuteri has been recognized as a homeostasis enhancer by promoting beneficial subgingival strains on major periodontal pathogens [15, 16]. In addition, and given that contradictory results refute the inflammatory or microbiological benefits of L. reuteri [17, 18], the present case report represents a research gap supporting monotherapy for the management of periodontitis, with the advantage of evaluating microbial levels and proportions with the Checkerboard technique of 40 representative periodontal strains.”

And, we added reference 18. https://doi.org/10.3109/00016357.2012.734406.

Point 4. The authors should elaborate on the etiopathogenesis of the periodontal disease and an emphasis on systemic inflammation. The authors can take a of reference https://doi.org/10.1038/s41577-020-00488-6, https://doi.org/10.3389/fphys.2021.709438.

Response 4: Following the reviewers' comments, we update information related to the etiopathogenesis of periodontal diseases and their relationship with systemic diseases (Page 1,2, lines 41 – 54).

“Periodontitis is an infectious and inflammatory disorder that is characterized by the destruction of the supporting structure of teeth, induced by biofilm dysbiosis, in which a higher proportion of major pathogens such as red complex species, including Porphyromonas gingivalis, Tannerella forsythia, and Treponema denticola, and a low proportion of beneficial strains, such as Actinomyces sp. or Streptococcus sp., remain the key to microbial disruption [3-5]. Despite improvements in preventive measures over the past 40 years, periodontal disease continues to contribute to widespread oral health dysfunction and an increased susceptibility to systemic diseases. Chronic unresolved hyperinflammation during periodontal disease is strongly associated with systemic conditions such as diabetes and obesity, and cardiovascular and neurological diseases, and is caused by the dysbiosis of the oral microbiome; therefore, major periodontal pathogens may be the targets of therapeutic interventions, as has recently been reported in the literature [6, 7].”

And, we added reference 7: https://doi.org/10.3389/fphys.2021.709438.

Point 5. The author should mention the research gap in the introduction. The authors have mentioned few studies in the discussion section. I would suggest this should be written in introduction section and state the novelty of the current study.

Response 5: Following the reviewers' comments, we added the following information (page 2, lines 75 – 79).

“In addition, and given that contradictory results refute the inflammatory or microbiological benefits of L. reuteri [17, 18], the present case report represents a research gap supporting monotherapy for the management of periodontitis, with the advantage of evaluating microbial levels and proportions with the Checkerboard technique of 40 representative periodontal strains.”

Material and method:

Point 1. Kindly mention the stage and grade of periodontitis patient.

Response 1: This information is presented on page 6, lines 205 – 206, as follows:

“….the patient was diagnosed with generalized periodontitis, stage IV, grade B [4]…”

Results:

Point 1. Clinical / microbiological pictures can be incorporated.

Response 1: At the reviewer's suggestion, representative clinical images were incorporated (Supplementary Figure 2). While the main microbiological findings are shown represented in Figures 1, 2, and 3.

Discussion:

Point 1. Kindly comment why the authors have chosen the study as a case report. Why they have not recruited more study samples to complete it as a original article or case series?

Response 1: It was decided to present this manuscript as a case report because we had the ideal conditions to test the efficacy of L. reuteri monotherapy, the patient was not willing to receive periodontal mechanical therapy, and she had not consumed antibiotics or received periodontal mechanical treatment in the past.

The present clinical case represents the first approach for our research group for future longitudinal clinical studies, based on previously reported literature [Claffey, N.; et al. Periodontology 2000 2004, 36, (1), 35-44.  Iniesta, M.; et al. A randomized clinical trial. Journal of clinical periodontology 2012, 39, (8), 736-44. Sinkiewicz, G.; et al. Swedish dental journal 2010, 34, (4), 197-206.], evaluating changes in the subgingival microbiota after the use of L. reuteri as monotherapy.

Point 2. Write limitations and future directions as separate headings

Response 2: Following the reviewer's suggestion, the following information was added on page 12 lines 417 - 421:

“Therefore, the present case report represents an approach toward future perspectives, including case-control investigations using the probiotic L. reuteri, with and without periodontal mechanical therapy, and the use of new genomic technologies such as 16S gene analysis or culturomics to identify a larger number of subgingival strains [29].”

Round 2

Reviewer 1 Report

Dear authors,

thanks for having enclosed all the changes suggested.

I think the manuscript is now publishable in its present form.

Regards